# A Review of Recent Advances in Brain Tumor Diagnosis Based on AI-Based Classification

**DOI:** 10.3390/diagnostics13183007

**Published:** 2023-09-20

**Authors:** Reham Kaifi

**Affiliations:** 1Department of Radiological Sciences, College of Applied Medical Sciences, King Saud bin Abdulaziz University for Health Sciences, Jeddah City 22384, Saudi Arabia; kaifir@ksau-hs.edu.sa; 2King Abdullah International Medical Research Center, Jeddah City 22384, Saudi Arabia; 3Medical Imaging Department, Ministry of the National Guard—Health Affairs, Jeddah City 11426, Saudi Arabia

**Keywords:** brain tumors, magnetic resonance imaging, computed tomography, computer-aided diagnostic and detection, deep learning, machine learning

## Abstract

Uncontrolled and fast cell proliferation is the cause of brain tumors. Early cancer detection is vitally important to save many lives. Brain tumors can be divided into several categories depending on the kind, place of origin, pace of development, and stage of progression; as a result, tumor classification is crucial for targeted therapy. Brain tumor segmentation aims to delineate accurately the areas of brain tumors. A specialist with a thorough understanding of brain illnesses is needed to manually identify the proper type of brain tumor. Additionally, processing many images takes time and is tiresome. Therefore, automatic segmentation and classification techniques are required to speed up and enhance the diagnosis of brain tumors. Tumors can be quickly and safely detected by brain scans using imaging modalities, including computed tomography (CT), magnetic resonance imaging (MRI), and others. Machine learning (ML) and artificial intelligence (AI) have shown promise in developing algorithms that aid in automatic classification and segmentation utilizing various imaging modalities. The right segmentation method must be used to precisely classify patients with brain tumors to enhance diagnosis and treatment. This review describes multiple types of brain tumors, publicly accessible datasets, enhancement methods, segmentation, feature extraction, classification, machine learning techniques, deep learning, and learning through a transfer to study brain tumors. In this study, we attempted to synthesize brain cancer imaging modalities with automatically computer-assisted methodologies for brain cancer characterization in ML and DL frameworks. Finding the current problems with the engineering methodologies currently in use and predicting a future paradigm are other goals of this article.

## 1. Introduction

The human brain, which serves as the control center for all the body’s organs, is a highly developed organ that enables a person to adapt to and withstand various environmental situations [1]. The human brain allows people to express themselves in words, carry out activities, and express thoughts and feelings. Cerebrospinal fluid (CSF), white matter (WM), and gray matter (GM) are the three major tissue components of the human brain. The gray matter regulates brain activity and comprises neurons and glial cells. The cerebral cortex is connected to other brain areas through white matter fibers comprising several myelinated axons. The corpus callosum, a substantial band of white matter fibers, connects the left and right hemispheres of the brain [2]. A brain tumor is a brain cell growth that is out of control and aberrant. Any unanticipated development may affect human functioning since the human skull is a rigid and volume-restricted structure, depending on the area of the brain involved. Additionally, it might spread to other organs, further jeopardizing human functions [3]. Early cancer detection makes the ability to plan effective treatment possible, which is crucial for the healthcare sector [4]. Cancer is difficult to cure, and the odds of survival are significantly reduced if it spreads to nearby cells. Undoubtedly, many lives could be preserved if cancer was detected at its earliest stage using quick and affordable diagnostic methods. Both invasive and noninvasive approaches may be utilized to diagnose brain cancer. An incision is made during a biopsy to extract a lesion sample for analysis. It is regarded as the gold standard for the diagnosis of cancer, where pathologists examine several cell characteristics of the tumor specimen under a microscope to verify the malignancy.

Noninvasive techniques include physical inspections of the body and imaging modalities employed for imaging the brain [5]. In comparison to brain biopsy, other imaging modalities, such as CT scans and MRI images, are more rapid and secure. Radiologists use these imaging techniques to identify brain problems, evaluate the development of diseases, and plan surgeries [6]. However, brain scans or image interpretation to diagnose illnesses are prone to inter-reader variability and accuracy, which depends on the medical practitioner’s competency [5]. It is crucial to accurately identify the type of brain disorder to reduce diagnostic errors. Utilizing computer-aided diagnostic (CAD) technologies can improve accuracy. The fundamental idea behind CAD is to offer a computer result as an additional guide to help radiologists interpret images and shorten the reading time for images. This enhances the accuracy and stability of radiological diagnosis [7]. Several CAT-based artificial intelligence techniques, such as machine learning (ML) and deep learning (DL), are described in this review for diagnosing tissues and segmenting tumors. The segmentation process is a crucial aspect of image processing. This approach includes a procedure for extracting the area that helps determine whether a region is infected. Using MRI images to segment brain tumors presents various challenges, including image noise, low contrast, loss borders, shifting intensities inside tissues, and tissue-type variation.

The most complex and crucial task in many medical image applications is detecting and segmenting brain tumors because it often requires much data and information. Tumors come in a variety of shapes and sizes. Automatic or semiautomatic detection/segmentation, helped by AI, is currently crucial in medical diagnostics. The medical professionals must authenticate the boundaries and areas of the brain cancer and ascertain where precisely it rests and the exact impacted locations before therapies such as chemotherapy, radiation, or brain surgery. This review examines the output from various algorithms that are used in segmenting and detecting brain tumors.

The review is structured as follows: Types of brain tumors are described in Section 2. The imaging modalities utilized in brain imaging are discussed in Section 3. The review algorithms used in the study are provided in Section 4. A review of the relevant state-of-the-art is provided in Section 5. The review is discussed in Section 6. The work’s conclusion is presented in Section 7. 

## 2. Types of Brain Tumors

The main three parts of the brain are the brain stem, cerebrum, and cerebellum [1]. The cerebellum is the second-largest component of the brain and manages bodily motor activities, including balance, posture, walking, and general coordination of movements. It is positioned behind the brain and connected to the brain stem. Internal white matter, tiny but deeply positioned volumes of gray matter, and a very thin gray matter outer cortex can all be found in the cerebellum and cerebrum. The brainstem links to the spinal cord. It is situated at the brain’s base. Vital bodily processes, including motor, sensory, cardiac, repositories, and reflexes, are all under the control of the brainstem. Its three structural components are the medulla oblongata, pons, and midbrain [2]. A brain tumor is the medical term for an unexpected growth of brain cells [8]. According to the tumor’s location, the kind of tissue involved, and whether they are malignant or benign, scientists have categorized several types of brain tumors based on the location of the origin (primary or secondary) and additional contributing elements [9]. The World Health Organization (WHO) categorized brain tumors into 120 kinds. This categorization is based on the cell’s origin and behavior, ranging from less aggressive to greater aggressive. Even certain tumor forms are rated, with grades I being the least malignant (e.g., meningiomas, pituitary tumors) and IV being the most malignant. Despite differences in grading systems that rely on the kind of tumor, this denotes the pace of growth [10]. The most frequent type of brain tumor in adults is glioma, which may be classified into HGG and LGG. The WHO further categorized LGG into I–II grade tumors and HGG into III–IV grade. To reduce diagnosing errors, accurate identification of the specific type of brain disorder is crucial for treatment planning. A summary of various types of brain tumors is provided in Table 1.

## 3. Imaging Modalities

For many years, the detection of brain abnormalities has involved the use of several medical imaging methods. The two brain imaging approaches are structural and functional scanning [11]. Different measurements relating to brain anatomy, tumor location, traumas, and other brain illnesses compose structural imaging [12]. The finer-scale metabolic alterations, lesions, and visualization of brain activity are all picked up by functional imaging methods. Techniques including CT, MRI, SPECT, positron emission tomography (PET), (FMRI), and ultrasound (US) are utilized to localize brain tumors for their size, location as well as shape, and other characteristics [13].

### 3.1. MRI

MRI is a noninvasive procedure that utilizes nonionizing, safe radiation [14] to display the 3D anatomical structure of any region of the body without the need for cutting the tissue. To acquire images, it employs RF pulses and an intense magnetic field [15].

The body is intended to be positioned within an intense magnetic field. The water molecules of the human body are initially in their equilibrium state when the magnets are off. The magnetic field is then activated by moving the magnets. The body’s water molecules align with the magnetic field’s direction under the effect of this powerful magnetic field [14]. Protons are stimulated to spin opposing the magnetic field and realign by the application of a high RF energy pulse to the body in the magnetic field’s direction. Protons are stimulated to spin opposing the magnetic field and realign by the application of a high RF energy pulse to the body in the magnetic field’s direction. When the RF energy pulse is stopped, the water molecules return to their state of equilibrium and align with the magnetic field once more [14]. This causes the water molecules to produce RF energy, which the scanner detects and transforms into visual images [16]. The tissue structure determines the amount of RF energy the water molecules can use. As we can see in Figure 1, healthy brain has white matter (WM), gray matter (GM), and CSF, according to a structural MRI scan [17]. The primary difference between these tissues in a structural MRI scan is based on the amount of water they contain, with WM constituting 70% water and GM containing 80% water. The CSF fluid is almost entirely composed of water, as shown in Figure 1. 

Figure 2 illustrates the fundamental MRI planes used to visualize the anatomy of the brain: axial, coronal, and sagittal. Tl, T2, and FLAIR MRI sequences are most often employed for brain analysis [14]. A Tl-weighted scan can distinguish between gray and white matter. T2-weighted imaging is water-content sensitive and is therefore ideally suited to conditions where water accumulates within the tissues of the brain. 

In pathology, FLAIR is utilized to differentiate between CSF and abnormalities in the brain. Gray-level intensity values in pixel spaces form an image during an MRI scan. The values of the gray-level intensity are dependent on the cell density. On T1 and T2 images of a tumor brain, the intensity level of the tumorous tissues differs [16]. The properties of various MRI sequences are shown in Table 2.

Most tumors show low or medium gray intensity on T1-w. On T2-w, most tumors exhibit bright intensity [17]. Examples of MRI tumor intensity level are shown in Figure 3.

Another type of MRI identified as functional magnetic resonance imaging (fMRI) [18] measures changes in blood oxygenation to interpret brain activity. An area of the brain that is more active begins to use more blood and oxygen. As a result, an fMRI correlates the location and mental process to map the continuing activity in the brain.

### 3.2. CT 

CT scanners provide finely detailed images of the interior of the body using a revolving X-ray beam and a row of detectors. On a computer, specific algorithms are used to process the images captured from various angles to create cross-sectional images of the entire body [19]. However, a CT scan can offer more precise images of the skull, spine, and other bone structures close to a brain tumor, as shown in Figure 4. Patients typically receive contrast injections to highlight aberrant tissues. The patient may occasionally take dye to improve their image. When an MRI is unavailable, and the patient has an implantation like a pacemaker, a CT scan may be performed to diagnose a brain tumor. The benefits of using CT scanning are low cost, improved tissue classification detection, quick imaging, and more widespread availability. The radiation risk in a CT scan is 100 times greater than in a standard X-ray diagnosis [19].

### 3.3. PET

An example of a nuclear medicine technique that analyzes the metabolic activity of biological tissues is positron emission tomography (PET) [20]. Therefore, to help evaluate the tissue being studied, a small amount of a radioactive tracer is utilized throughout the procedure. Fluorodeoxyglucose (FDG) is a popular PET agent for imaging the brain. To provide more conclusive information on malignant (cancerous) tumors and other lesions, PET may also be utilized in conjunction with other diagnostic procedures like CT or MRI. PET scans an organ or tissue by utilizing a scanning device to find photons released by a radionuclide at that site [20]. The chemical compounds that are normally utilized by the specific organ or tissue throughout its metabolic process are combined with a radioactive atom to create the tracer used in PET scans, as shown in Figure 5.

### 3.4. SPECT 

A nuclear imaging examination called a single-photon emission computed tomography (SPECT) combines CT with a radioactive tracer. The tracer is what enables medical professionals to observe the blood flow to tissues and organs [21]. A tracer is injected into the patient’s bloodstream prior to the SPECT scan. The radiolabeled tracer generates gamma rays that the CT scanner can detect since it is radiolabeled. Gamma-ray information is gathered by the computer and shown on the CT cross-sections. A 3D representation of the brain can be created by adding these cross-sections back together [21].

### 3.5. Ultrasound 

An ultrasound is a specialized imaging technique that provides details that can be useful in cancer diagnosis, especially for soft tissues. It is frequently employed as the initial step in the typical cancer diagnostic procedure [22]. One advantage of ultrasound is that a test can be completed swiftly and affordably without subjecting the patient to radiation. However, ultrasound cannot independently confirm a cancer diagnosis and is unable to generate images with the precise level of resolution or detail like a CT or MRI scan. A medical expert gently moves a transducer throughout the patient’s skin across the region of the body being examined during a conventional ultrasound examination. A succession of high-frequency sounds is generated by the transducer, which “bounce off” the patient’s interior organs. The ensuing echoes return to the ultrasound device, which then transforms the sound waves into a 2D image that may be observed in real-time on a monitor. According to [22], US probes have been applied in brain tumor resection. According to the degree of density inside the tissue being assessed, the shape and strength of ultrasonic echoes can change. An ultrasound can detect tumors that may be malignant because solid masses and fluid-filled cysts bounce sound waves differently.

## 4. Classification and Segmentation Method

As was stated in the introduction, brain tumors are a leading cause of death worldwide. Computer-aided detection and diagnosis refer to software that utilizes DL, ML, and computer vision for analyzing radiological and pathological images. It has been created to assist radiologists in diagnosing human disease in various body regions, including applications for brain tumors. This review explored different CAT-based artificial intelligence approaches, including ML and DL, for automatically classifying and segmenting tumors.

### 4.1. Classification Methods

A classification is an approach in which related datasets are grouped together according to common features. A classifier in classification is a model created for predicting the unique features of a class label. Predicting the desired class for each type of data is the fundamental goal of classification. Deep learning and machine learning techniques are used for the classification of medical images. The key distinction between the two types is the approach for obtaining the features used in the classification process.

#### 4.1.1. Machine Learning

ML is a branch of AI that allows computers to learn without being explicitly programmed. Classifying medical images, including lesions, into various groups using input features has become one of the latest applications of ML. There are two types of ML algorithms: supervised learning and unsupervised learning [23]. ML algorithms learn from labeled data in supervised learning. Unsupervised learning is the process by which ML systems attempt to comprehend the interdata relationship using unlabeled data. ML has been employed to analyze brain cancers in the context of brain imaging [24]. The main stages of ML classification are image preprocessing, feature extraction, feature selection, and classification. Figure 6 illustrates the process architecture.

1.Data Acquisition

As previously noted, we can collect brain cancer images using several imaging modalities such as MRI, CT, and PET. This technique effectively visualizes aberrant brain tissues. 

2.Preprocessing

Preprocessing is a very important stage in the medical field. Normally, noise enhancement or reduction in images occurs during preprocessing. Medical noise significantly reduces image quality, making them diagnostically inefficient. To properly classify medical images, the preprocessing stage must be effective enough to eliminate as much noise as possible without affecting essential image components [25]. This procedure is carried out using a variety of approaches, including cropping, image scaling, histogram equalization, filtering using a median filter, and image adjusting [26].

3.Feature extraction

The process of converting images into features based on several image characteristics in the medical field is known as feature extraction. These features carry the same information as the original images but are entirely different. This technique has the advantages of enhancing classifier accuracy, decreasing overfitting risk, allowing users to analyze data, and speeding up training [27]. Texture, contrast, brightness, shape, gray level co-occurrence matrix (GLCM) [28], Gabor transforms [29], wavelet-based features [30], 3D Haralick features [31], and histogram of local binary patterns (LBP) [32] are some of the examples of the various types of features.

4.Feature selection

The technique attempts to arrange the features in ascending order of importance or relevance, with the top features being mostly employed in classification. As a result, multiple feature selection techniques are needed to reduce redundant information to discriminate between relevant and nonrelated features [33], such as PCA [34], genetic algorithm (GA) [35], and ICA [36].

5.ML algorithm

Machine learning aims to divide the input information into separate groups based on common features or patterns of behavior. KNN [35], ANN [37], RF [38], and SVM [39] are examples of supervised methods. These techniques include two stages: training and testing. During training, the data are manually labeled using human involvement. The model is first constructed in this step, after which it is utilized to determine the classes that are unlabeled in the testing stage. Application of the KNN algorithm works by finding the points that are closest to each other by computing the distance between them using one of several different approaches, including the Hamming, Manhatten, Euclidean, and Minkowski distances [35].

The support vector machine (SVM) technique is frequently employed for classification tasks. Every feature forming a data point in this approach, which represents a coordinate, is formed in a distinct n-space. As a result, the objective of the SVM method is to identify a boundary or line across a space with n dimensions, referred to as a hyperplane that separates classes [39]. There are numerous ways to create different hyperplanes, but the one with the maximum margin is the best. The maximum margin is the separation between the most extreme data points inside a class, often known as the support vectors.

#### 4.1.2. Extreme Learning Machine (ELM)

Another new field that uses less computing than neural networks is evolutionary machine learning (EML). It is based on the real-time classification and regression technique known as the single-layer feed-forward neural network (SLFFNN). The input-to-hidden layer weights in the ELM are initialized randomly, whereas the hidden-to-output layer weights are trained to utilize the Moore–Penrose inverse method [40] to obtain a least-squares solution. As a result, classification accuracy is increased while net complexity, training time, and learning speed are all reduced.

Additionally, the hidden layer weights provide the network the capacity to multitask similar to other ML techniques such as KNN, SVM, and Bayesian networks [40]. As shown in Figure 7, the ELM network is composed of three levels, all of which are connected. Weights between the hidden and output layers can only vary, but the weights between the input and hidden layers are initially fixed at random and remain so during training.

#### 4.1.3. Deep Learning (DL)

Beginning a few years ago, deep learning, a branch of machine learning, has been utilized extensively to create automatic, semiautomatic, and hybrid models that can accurately detect and segment tumors in the shortest period possible [41]. DL can learn the features that are significant for a problem by utilizing a training corpus with sufficient diversity and quality. Deep learning [42] has achieved excellent success in tackling the issues of ML by combining the feature extraction and selection phases into the training process [43]. Deep learning is motivated by the comprehension of neural networks that exist within the human brain. DL models are often represented as a sequence of layers generated by a weighted sum of information from the previous layer. The data are represented by the first layer, while the output is represented by the last layer [44]. Deep learning models can tackle extremely difficult problems while often requiring less human interaction than conventional ML techniques because several layers make it possible to duplicate complex mapping functions.

The most common DL model used for the categorization and segmentation of images is a convolution neural network (CNN). In a hierarchical manner, CNN analyzes the spatial relationship of pixels. Convoluting the images with learned filters creates a hierarchy of feature maps, which is how this is accomplished. This convolution function is performed in several layers such that the features are translation- and distortion-invariant and hence accurate to a high degree [45]. Figure 8 illustrates the main process in DL.

Preprocessing is primarily used to eliminate unnecessary variation from the input image and make training the model easier. More actions are required to extend beyond neural network models’ limits, such as resizing normalization. All images must be resized before being entered into CNN classification models since DL requires inputs of a constant size [46]. Images that are greater than the desired size can be reduced by downscaling, interpolation, or cutting the background pixels [46].

Many images are required for CNN-based classification. Data augmentation is one of the most important data strategies for addressing issues with unequal distribution and data paucity [47].

CNN’s architecture is composed of three primary layers: convolutional, pooling, and fully connected. The first layer is the main layer that is able to extract image features such as edges and boundaries. Based on the desired prediction results, this layer may automatically learn many filters in parallel for the training dataset. The first layer creates features, but the second layer oversees data reduction, which minimizes the size of those features and reduces the demand for computing resources. Every neuron in the final layer, which is a completely connected layer, is coupled to every neuron in the first layer. The layer serves as a classifier to classify the acquired feature vector of previous layers [48,49]. The approach that CNN uses is similar to how various neural networks work: it continually modifies its weights by taking an error from the output and inserting it as output to improve filters and weights. In addition, CNN standardizes the output utilizing a SoftMax function [50]. Many types of CNN architecture exist, including ResNet, AlexNet, and cascade-CNN, among others [51].

### 4.2. Segmentation Method

Brain tumor segmentation, which has been employed in some research, is an important step in improving disease diagnosis, evaluation, treatment plans, and clinical trials. The purpose of segmentation in tumor classification is to detect the tumor location from brain scans, improve representation, and allow quantitative evaluations of image structures during the feature extraction step [52]. Brain tumor segmentation can be accomplished in two ways: manually and completely automatically [53].

Manual tumor segmentation from brain scans is a difficult and time-consuming procedure. Furthermore, the artifacts created during the imaging procedure result in poor-quality images that are difficult to analyze. Additionally, due to uneven lesions, geographical flexibility, and unclear borders, manual detection of brain tumors is challenging. This section discusses several automated brain tumor segmentation strategies to help radiologists overcome these issues.

#### 4.2.1. Region-Based Segmentation

A region in an image is a collection of related pixels that comply with specific homogeneity requirements, such as shape, texture, and pixel intensity values [54]. In a region-based segmentation, the image is divided into disparate areas to precisely identify the target region [55]. When grouping pixels together, the region-based segmentation takes into consideration the pixel values, such as gray-level variance and difference, as well as their spatial closeness, such as the Euclidean distance or region density. K-means clustering [56] and FCM [56] are the most techniques used in this method.

#### 4.2.2. Thresholding Methods

The thresholding approach is a straightforward and effective way to separate the necessary region [57], but finding an optimum threshold in low-contrast images may be challenging.

Based on picture intensity, threshold values are chosen using histogram analysis [58]. There are two types of thresholding techniques: local and global. The global thresholding approach is the best choice for segmentation if the objects and the background have highly uniform brightness or intensity. The Gaussian distribution approach may be used to obtain the ideal threshold value [59]. Otsu thresholding [38] is the popular method among these techniques. 

#### 4.2.3. Watershed Techniques

The intensities of the image are analyzed using watershed techniques [60]. Topological watershed [61], marker-based watershed [62], and image IFT watershed [63] are a few examples of watershed algorithms.

#### 4.2.4. Morphological-Based Method

The morphology technique relies on the morphology of image features. It is mostly used for extracting details from images based on shape representation. Dougherty [64] defines dilation and erosion as two basic operations. Dilation is used to increase the size of an image. Erosion reduces the size of images.

#### 4.2.5. Edge-Based Method

Edge detection is performed using variations in image intensity. Pixels at an edge are those where the image’s function abruptly changes. Edge-based segmentation techniques include those by Sobel, Roberts, Prewitt, and Canny [65]. Reference [66] offers an enhanced edge detection approach for tumor segmentation. The development of an automated image-dependent thresholding is combined with the Sobel operator to identify the edges of the brain tumor.

#### 4.2.6. Neural-Networks-Based Method

Neuronal network-based segmentation techniques employ computer models of artificial neural networks consisting of weighted connections between processing units (called neurons). At the connections, the weights act as multipliers. To acquire the coefficient values, training is necessary. The segmentation of medical images and other fields has made use of a variety of neural network designs. Some of the techniques utilized in the segmentation process include the multilayer perceptron (MLP), Hopfield neural networks (HNN) [67], back-propagation learning algorithm, SVM-based segmentation [68], and self-organizing maps (SOM) neural network [67].

#### 4.2.7. DL-Based Segmentation 

The primary strategy used in the DL-based segmentation of brain tumors technique is to pass an image through a series of deep learning structures before performing input image segmentation based on the deep features [69]. Many deep learning methods, such as deep CNNs, CNN, and others, have been suggested for segmenting brain tumors.

A deep learning system called semantic segmentation [70] arranges pixels in an image according to semantic categories. The objective is to create a dense pixel-by-pixel segmentation map of the image, and each pixel is given an assigned category or entity.

### 4.3. Performance Evaluation 

An important component of every research work involves evaluating the classification and segmentation performance. The primary goal of this evaluation is to measure and analyze the model’s capability for segmentation or diagnostic purposes. Segmentation is a crucial step in improving the diagnostic process, as we mentioned before, but for this to occur, the segmentation process must be as accurate as feasible. Additionally, to evaluate the diagnostic approach utilized while taking complexity and time into account [71].

True positive (TP), true negative (TN), false positive (FP), and false negative (FN) are the main four elements in any analysis or to evaluate any segmentation or classification algorithm. A pixel that is accurately predicted to be assigned to the specified class in a segmentation method is represented by TP and TN based on the ground truth. Furthermore, FP is a result when the model predicts a pixel wrongly as not belonging to a specific class. A false negative (FN) results when the model wrongly predicts a pixel belonging to a certain class [71].

TP in classification tasks refers to an image that is accurately categorized into a positive category based on the ground truth. Similar to this, the TN result occurs when the model properly classifies an image in the negative category. As opposed to that, FP results occur when the model wrongly assigns an image in the positive class while the actual datum is in the negative category. FN results occur when the model misclassifies an image while it belongs in the positive category. Through the four elements mentioned above, different performance measures enable us to expand the analysis.

Accuracy (ACC) measures a model’s ability to correctly categorize all pixels/classes, whether they are positive or negative. Sensitivity (SEN) shows the percentage of accurately predicted positive images/pixels among all actual positive samples. It evaluates a model’s ability to recognize relevant samples or pixels. The percentage of actual negatives that were predicted is known as specificity (SPE). It indicates a percentage of classes or pixels that could not be accurately recognized [71].

The precision (PR) or positive predictive value (PPV) measures how frequently the model correctly predicts the class or pixel. It provides the precise percentage of positively expected results from models. The most often used statistic that combines SEN and precision is the F1 score [72]. It refers to the two-dimensional harmonic mean.

The Jaccard index (JI), also known as intersection over union (IoU), calculates the percentage of overlap between the model’s prediction output and the annotation ground-truth mask.

The spatial overlap between the segmented region of the model and the ground-truth tumor region is measured by the Dice similarity coefficient (DSC). A DSC value of zero means there is no spatial overlap between the annotated model result and the actual tumor location, whereas a value of one means there is complete spatial overlap. The receiver characteristics curve is summarized by the area under the curve (AUC), which compares SEN to the false positive rate as a measure of a classifier’s ability to discriminate between classes.

The similarity between the segmentation produced by the model and the expert-annotated ground truth is known as the similarity index (SI). It describes how the identification of the tumor region is comparable to that of the input image [71]. Table 3 summarizes different performance equations. 

## 5. Literature Review

### 5.1. Article Selection

The major goal of this study is to review and understand brain tumor classification and detection strategies developed worldwide between 2010 and 2023. This present study aims to review the most popular techniques for detecting brain cancer that have been made available globally, in addition to looking at how successful CAD systems are in this process.

We did not target any one publisher specifically, but we utilized articles from a variety of sources to account for the diversity of knowledge in a particular field. We collected appropriate articles from several internet scientific research article libraries. We searched the pertinent publications using IEEE Explore, Medline, ScienceDirect, Google Scholar, and ResearchGate.

Each time, the filter choice for the year (2010 to 2023) was chosen so that only papers from the chosen period were presented. Most frequently, we used terms like “detection of MRI images using deep learning,” “classification of brain tumor from CT/MRI images using deep learning,” “detection and classification of brain tumor using deep learning,” “CT brain tumor,” “PET brain tumor,” etc. This study offers an analysis of 53 chosen publications. 

### 5.2. Publicly Available Datasets

The researchers tested the proposed methods on several publicly accessible datasets. In this part, several significant and difficult datasets are covered. The most difficult MRI datasets are BRATS. Table 4 presents a summary of the dataset names.

### 5.3. Related Work

In addition to the several techniques for segmenting brain tumors that we already highlighted, this section presents a summary of studies that use artificial intelligence to classify brain tumors.

#### 5.3.1. MRI Brain Tumor Segmentation

This section describes the various machine learning, deep learning, region growth, thresholding, and literature-proposed brain tumor segmentation strategies.

To segment brain tumors, Gordillo et al. [80] utilized fuzzy logic structure, which they built utilizing features extracted from MR images and expert knowledge. This system learns unsupervised and is fully automated. With trials conducted on two different forms of brain tumors, glioblastoma multiform and meningioma, the result of segmentation using this approach is shown to be satisfactory, with the lowest accuracy of 71% and a maximum of 93%.

Employing fuzzy c-means clustering on MRI, Rajendran [81] presented logic analyzing for segmenting brain tumors. The region-based technique that iteratively progresses toward the ultimate tumor border was initialized using the tumor type output of fuzzy clustering. Using 15 MR images with manual segmentation ground truth available, tests were conducted on this approach to determine its effectiveness. The overall result was suitable, with a sensitivity of 96.37% and an average Jaccard coefficient value of 83.19%.

An SVM classifier was applied by Kishore et al. to categorize tumor pixels using vectors of features from MR images, such as mean intensity and LBP. Level sets and region-growing techniques were used for the segmentation. The experiments on their suggested methods used MR images with tumor regions manually defined by 11 different participants. Their suggested methods are effective, with a DSC score of 0.69 [82].

A framework for segmenting tumorous MRI 3D images was presented by Abbasi and Tajeripour [38]. The first phase improves the input image’s contrast using bias field correction. The data capacity is reduced using the multilevel Otsu technique in the second phase. LBP in three orthogonal planes and an enhanced histogram of images are employed in the third stage, the feature extraction step. Lastly, the random forest is employed as a classifier for distinguishing tumorous areas since it can work flawlessly with large inputs and has a high level of segmentation accuracy. The overall outcome was acceptable, with a mean Jaccard value of 87% and a DSC of 93%.

By combining two K-means and FCM-clustering approaches, Almahfud et al. [83] suggest a technique for segmenting human brain MRI images to identify brain cancers. Because K-means is more susceptible to color variations, it can rapidly and effectively discover optima and local outliers. So that the cluster results are better and the calculation procedure is simpler, the K-means results are clustered once more with FCM to categorize the convex contour based on the border. To increase accuracy, morphology and noise reduction procedures are also suggested. Sixty-two brain MRI scans were used in the study, and the accuracy rate was 91.94%.

According to Pereira et al. [69], an automated segmentation technique based on CNN architecture was proposed, which explores small three-by-three kernels. Given the smaller number of weights in the network, using small kernels enables the creation of more intricate architectures and helps prevent overfitting. Additionally, they looked at the use of intensity normalizing as an initial processing step, which, when combined with data augmentation, was highly successful in segmenting brain tumors in MRI images. Their suggestion was verified using the BRATS database, yielding Dice similarity coefficient values of 0.88, 0.83, and 0.77 for the Challenge dataset for the whole, core, and enhancing areas.

According to the properties of a separated local square, they suggested a unique approach for segmenting brain tumors [84]. The suggested procedure essentially consists of three parts. An image was divided into homogenous sections with roughly comparable properties and sizes using the super-pixel segmentation technique in the first stage. The second phase was the extraction of gray statistical features and textural information. In the last phase of building the segmentation model, super-pixels were identified as either tumor areas or nontumor regions using SVM. They used 20 images from the BRATS dataset, where a DSC of 86.12% was attained, to test the suggested technique.

The CAD system suggested by Gupta et al. [85] offers a noninvasive method for the accurate tumor segmentation and detection of gliomas. The system takes advantage of the super pixels’ combined properties and the FCM-clustering technique. The suggested CAD method recorded 98% accuracy for glioma detection in both low-grade and high-grade tumors.

Brain tumor segmentation using the CNN-based data transfer to SVM classifier approach was proposed by Cui et al. [68]. Two cascaded phases comprise their algorithm. They trained CNN in the initial step to understand the mapping of the image region to the tumor label region. In the testing phase, they passed the testing image and CNN’s anticipated label output to an SVM classifier for precise segmentation. Tests and evaluations show that the suggested structure outperforms separate SVM-based or CNN-based segmentation, while DSC achieved 86.12%.

The two-pathway-group CNN architecture described by Razzak et al. is a novel approach for brain tumor segmentation that simultaneously takes advantage of local and global contextual traits. This approach imposes equivariance in the 2PG-CNN model to prevent instability and overfitting parameter sharing. The output of a basic CNN is handled as an extra source and combined at the last layer of the 2PG CNN, where the cascade architecture was included. When a group CNN was embedded into a two-route architecture for model validation using BRATS datasets, the results were DSC 89.2%, PR 88.22%, and SEN 88.32% [86].

A semantic segmentation model for the segmentation of brain tumors from multimodal 3D MRIs for the BRATS dataset was published in [87]. After experimenting with several normalizing techniques, they discovered that group-norm and instance-norm performed equally well. Additionally, they have tested with more advanced methods of data augmentation, such as random histogram pairing, linear image transformations, rotations, and random image filtering, but these have yet to show any significant benefit. Further, raising the network depth had no positive effect on performance. However, increasing the number of filters consistently produced better results. Their BRATS end testing dataset values were 0.826, 0.882, and 0.837 for overall Dice coefficient or improved tumor core, entire tumor, and tumor center, respectively.

CNN was used by Karayegen and Aksahin [88] to offer a semantic segmentation approach for autonomously segmenting brain tumors on BRATS image datasets that include images from four distinct imaging modalities (T1, T1C, T2, and FLAIR). This technique was effectively used, and images were shown in a variety of planes, including sagittal, coronal, and axial, to determine the precise tumor location and parameters such as height, breadth, and depth. In terms of tumor prediction, evaluation findings of semantic segmentation carried out using networks are incredibly encouraging. The mean IoU and mean prediction ratio were both calculated to be 86.946 and 91.718, respectively.

A novel, completely automatic method for segmenting brain tumor regions was proposed by Ullah et al. [89] using multiscale residual attention CNN (MRA-UNet). To maintain the sequential information, MRA-UNet uses three sequential slices as its input. By employing multiscale learning in a cascade path, it can make use of the adaptable region of interest strategy and precisely segment improved and core tumor regions. In the BRATS-2020 dataset, their method produced novel outcomes with an overall Dice score of 90.18%.

A new technique for segmenting brain tumors using the fuzzy Otsu thresholding morphology (FOTM) approach was presented by Wisaeng and Sa-Ngiamvibool [90]. The values from each single histogram in the original MRI image were modified by using a color normalizing preprocessing method in conjunction with histogram specification. The findings unambiguously demonstrate that image gliomas, image meningiomas, and image pituitary have average accuracy indices of 93.77%, 94.32%, and 94.37%, respectively. A summary of MRI brain tumor segmentation is provided in Table 5.

#### 5.3.2. MRI Brain Tumor Classification Using ML

The automated classification of brain cancers using MRI images has been the subject of several studies. Cleaning data, feature extraction, and feature selection are the basic steps in the machine learning (ML) process that have been used for this purpose. Building an ML model based on labeled samples is the last step. A summary of MRI brain tumor classification using ML is provided in Table 6.

An NN-based technique to categorize a given MR brain image as either normal or abnormal is presented in [91]. In this method, features were first extracted from images using the wavelet transform, and then the dimensionality of the features was reduced using PCA methodology. The reduced features were routed to a back-propagation NN that uses a scaled conjugate gradient (SCG) to determine the best weights for the NN. This technique was used on 66 images, 18 of which were normal and 48 abnormal. On training and test images, the classification accuracy was 100%.

An automated and efficient CAD method based on ensemble classifiers was proposed by Arakeri and Reddy [36] for the classification of brain cancers on MRI images as benign or malignant. A tumor’s texture, shape, and border properties were extracted and used as a representation. The ICA approach was used to select the most significant features. The ensemble classifier, consisting of SVM, ANN, and kNN classifiers, is trained using these features to describe the tumor. A dataset consisting of 550 patients’ T1- and T2-weighted MR images was used for the experiments. With an accuracy of 99.09% (sensitivity 100% and specificity 98.21%), the experimental findings demonstrated that the suggested classification approach achieves strong agreement with the combined classifier and is extremely successful in the identification of brain tumors. Figure 9 illustrates the CAD method based on ensemble classifiers.

In [92], the authors suggested a novel, wavelet-energy-based method for automatically classifying MR images of the human brain into normal or abnormal. The classifier was SVM, and biogeography-based optimization (BBO) was utilized to enhance the SVM’s weights. They succeeded in achieving 99% precision and 97% accuracy.

Amin et al. [28] suggest an automated technique to distinguish between malignant and benign brain MRI images. The segmentation of potential lesions has used a variety of methodologies. Then, considering shape, texture, and intensity, a feature set was selected for every candidate lesion. The SVM classifier is then used on the collection of features to compare the proposed framework’s precision using various cross-validations. Three benchmark datasets, including Harvard, Rider, and Local, are used to verify the suggested technique. For the procedure, the average accuracy was 97.1%, the area under the curve was 0.98, the sensitivity was 91.9%, and the specificity was 98.0%.

A suitable CAD approach toward classifying brain tumors is proposed in [93]. The database includes meningioma, astrocytoma, normal brain areas, and primary brain tumors. The radiologists selected 20 × 20 regions of interest (ROIs) for every image in the dataset. Altogether, these ROI(s) were used to extract 371 intensity and texture features. These three classes were divided using the ANN classifier. Overall classification accuracy was 92.43%.

Four hundred twenty-eight T1 MR images from 55 individuals were used in a varied dataset for multiclass brain tumor classification [94]. A based-on content active contour model extracted 856 ROIs. These ROIs were used to extract 218 intensity and texture features. PCA was employed in this study to reduce the size of the feature space. The ANN was then used to classify these six categories. The classification accuracy was seen to have reached 85.5%.

A unique strategy for classifying brain tumors in MRI images was proposed in [95] by employing improved structural descriptors and hybrid kernel-SVM. To better classify the image and improve the texture feature extraction process using statistical parameters, they used GLCM and histograms to derive the texture feature from every region. Different kernels were combined to create a hybrid kernel SVM classifier to enhance the classification process. They applied this technique to only axial T1 brain MRI images—93% accuracy for their suggested strategy.

A hybrid system composed of two ML techniques was suggested in [96] for classifying brain tumors. For this, 70 brain MR images overall (60 abnormal, 10 normal) were taken into consideration. DWT was used to extract features from the images. Using PCA, the total number of features was decreased. Following feature extraction, feed-forward back-propagation ANN and KNN were applied individually on the decreased features. The back-propagation learning method for updating weights is covered by FP-ANN. KNN has already been covered. Using KNN and FP-ANN, this technique achieves 97% and 98% accuracy, respectively [96]. 

A strategy for classifying brain MRI images was presented in [97]. Initially, they used an enhanced image improvement method that comprises two distinct steps: noise removal and contrast enhancement using histogram equalization. Then, using a DWT to extract features from an improved MR brain image, they further decreased these features by mean and standard deviation. Finally, they developed a sophisticated deep neural network (DNN) to classify the brain MRI images as abnormal or normal, and their strategy achieved 95.8%.

**Table 6 diagnostics-13-03007-t006:** MRI brain tumor classification using ML.

Ref.	Scan	Year	Feature Extraction	Feature Selection	Classification	Acc.
[96]	MRI	2010	GLCM	PCA	ANN and KNN	98% and 97%
[91]	MRI	2011	Wavelet	PCA	Back-propagation NN	100.00%
[94]	MRI	2013	Intensity and texture	PCA	ANN	85.50%
[95]	MRI	2014	GLCM	-	SVM	93.00%
[36]	MRI	2015	Texture and shape	ICA	SVM	99.09%
[92]	MRI	2015	Wavelet	-	SVM	97.00%
[28]	MRI	2017	Texture and shape	-	SVM	97.10%
[93]	MRI	2017	Intensity and texture	-	ANN	92.43%
[97]	MRI	2020	DWT	Mean and standard deviation	DNN	95.8%

#### 5.3.3. MRI Brain Tumor Classification Using DL

Difficulties remain in categorizing brain cancers from an MRI scan, despite encouraging developments in the field of ML algorithms for the classification of brain tumors into their different types. These difficulties are mostly the result of the ROI detection; typical labor-intensive feature extraction methods could be more effective [98]. Owing to the nature of deep learning, the categorization of brain tumors is now a data-driven problem rather than a challenge based on manually created features [99]. CNN is one of the deep learning models that is frequently utilized in brain tumor classification tasks and has produced a significant result [100].

According to a study [101], the CNN algorithm can be used to divide the severity of gliomas into two categories: low severity or high severity, as well as multiple grades of severity (Grades II, III, and IV). Accuracy rates of 71% and 96% were reached by the classifier.

A DL approach based on a CNN was proposed by Sultan et al. [7] to classify different kinds of brain tumors using two publicly available datasets. The proposed method’s block diagram is presented in Figure 10. The first divides cancers into meningioma, pituitary, and glioma tumors. The other one distinguishes among Grade II, III, and IV gliomas. The first and second datasets, which each have 233 and 73 patients, contain a combined total of 3064 and 516 T1 images. The suggested network configuration achieves the best overall accuracy, 96.13% and 98.7%, for the two studies, which results in significant performance [7].

Similarly, ref. [102] showed how to classify brain MRI scan images into malignant and benign using CNN algorithms in conjunction with augmenting data and image processing. They evaluated the effectiveness of their CNN model with pretrained VGG-16, Inception-v3, and ResNet-50 models using the transfer learning methodology. Even though the experiment was carried out on a relatively small dataset, the results reveal that the model’s accuracy result is quite strong and has a very low complexity rate, as it obtained 100% accuracy, compared to VGG-16’s 96%, ResNet-50’s 89%, and Inception-V3’s 75%. The structure of the suggested CNN architecture is shown in Figure 11.

For accurate glioma grade prediction, researchers developed a customized CNN-based deep learning model [103] and evaluated the performance using AlexNet, GoogleNet, and SqueezeNet by transfer learning. Based on 104 clinical glioma patients with (50 LGGs and 54 HGGs), they trained and evaluated the models. The training data was expanded using a variety of data augmentation methods. A five-fold cross-validation procedure was used to assess each model’s performance. According to the study’s findings, their specially created deep CNN model outperformed the pretrained models by an equal or greater percentage. The custom model’s accuracy, sensitivity, F1 score, specificity, and AUC values were, respectively, 0.971, 0.980, 0.970, 0.963, and 0.989.

A novel transfer learning-based active learning paradigm for classifying brain tumors was proposed by Ruqian et al. [104]. Figure 12 describes the workflow for active learning. On the MRI training dataset of 203 patients and the baseline validation dataset of 66 patients, they used a 2D slice-based technique to train and fine-tune the model. Their suggested approach allowed the model to obtain an area under the curve (ROC) of 82.89%. The researchers built a balanced dataset and ran the same process on it to further investigate the robustness of their strategy. Compared to the baseline’s AUC of 78.48%, the model’s AUC was 82%.

A total of 131 patients with glioma were enrolled [105]. A rectangular ROI was used to segment tumor images, and this ROI contained around 80% of the tumor. The test dataset was then created by randomly selecting 20% of the patient-level data. Models previously trained on the expansive natural image database ImageNet were applied to MRI images, and then AlexNet and GoogleNet were developed from scratch and fine-tuned. Five-fold cross-validation (CV) was used on the patient-level split to evaluate the classification task. The averaged performance metrics for validation accuracy, test accuracy, and test AUC from the five-fold CV of GoogleNet were, respectively, 0.867, 0.909, and 0.939.

Hamdaoui et al. [106] proposed an intelligent medical decision-support system for identifying and categorizing brain tumors using images from the risk of malignancy index. They employed deep transfer learning principles to avoid the scarcity of training data required to construct the CNN model. For this, they selected seven CNN architectures that had already been trained on an ImageNet dataset that they carefully fitted on (MRI) data of brain tumors gathered from the BRATS database, as shown in Figure 13. Just the prediction that received the highest score among the predictions made by the seven pretrained CNNs is produced to increase their model’s accuracy. They evaluated the effectiveness of the primary two-class model, which includes LGG and HGG brain cancers, using a ten-way cross-validation method. The test precision, F1 score, test precision, and test sensitivity for their suggested model were 98.67%, 98.06%, 98.33%, and 98.06%, respectively.

A new AI diagnosis model called EfficientNetB0 was created by Khazaee et al. [107] to assess and categorize human brain gliomas utilizing sequences from MR images. They used a common dataset (BRATS-2019) to validate the new AI model, and they showed that the AI components—CNN and transfer learning—provided outstanding performance for categorizing and grading glioma images, with 98.8% accuracy.

In [70], the researchers developed a model using transfer learning and pretrained ResNet18 to identify basal ganglia germinomas more accurately. In this retrospective analysis, 73 patients with basal ganglioma were enrolled. Based on both T1 and T2 data, brain tumors were manually segmented. To create the tumor classification model, the T1 sequence was utilized. Transfer learning and a 2D convolutional network were used. Five-fold cross-validation was used to train the model, and it resulted in a mean AUC of 88%.

Researchers suggested an effective hyperparameter optimization method for CNN based on Bayesian optimization [108]. This method was assessed by categorizing 3064 T1 images into three types of brain cancers (glioma, pituitary, and meningioma). Five popular deep pretrained models are compared to the improved CNN’s performance using transfer learning. Their CNN achieved 98.70% validation accuracy after applying Bayesian optimization.

A novel generated transfer DL model was developed by Alanazi et al. [109] for the early diagnosis of brain cancers into their different categories, such as meningioma, pituitary, and glioma. Several layers of the models were first constructed from scratch to test the performance of standalone CNN models performed for brain MRI images. The weights of the neurons were then revised using the transfer learning approach to categorize brain MRI images into tumor subclasses using the 22-layer, isolated CNN model. Consequently, the transfer-learned model that was created had an accuracy rate of 95.75%.

Rizwan et al. [110] suggested a method to identify various BT classes using Gaussian-CNN on two datasets. One of the datasets is employed to categorize lesions into pituitary, glioma, and meningioma. The other distinguishes between the three glioma classes (II, III, and IV). The first and second datasets, respectively, have 233 and 73 victims from a total of 3064 and 516 images on T1 enhanced images. For the two datasets, the suggested method has an accuracy of 99.8% and 97.14%.

A seven-layer CNN was suggested in [111] to assist with the three-class categorization of brain MR images. To decrease computing time, separable convolution was used. The suggested separable CNN model achieved 97.52% accuracy on a publicly available dataset of 3064 images. 

Several pretrained CNNs were utilized in [112], including GoogleNet, Alexnet, Resnet50, Resnet101, VGG-16, VGG-19, InceptionResNetV2, and Inceptionv3. To accommodate additional image categories, the final few layers of these networks were modified. Data from the clinical, Harvard, and Figshare repositories were widely used to assess these models. The dataset was divided into training and testing halves in a 60:40 ratio. The validation on the test set demonstrates that, compared to other proposed models, the Alexnet with transfer learning demonstrated the best performance in the shortest time. The suggested method obtained accuracies of 100%, 94%, and 95.92% using three datasets and is more generic because it does not require any manually created features.

The suggested framework [113] describes three experiments that classified brain malignancies such as meningiomas, gliomas, and pituitary tumors using three designs of CNN (AlexNet, VGGNet, and GoogleNet). Using the MRI slices of the brain tumor dataset from Figshare, each study then investigates transfer learning approaches like fine-tuning and freezing. The data augmentation approaches are applied to the MRI slices for results generalization, increasing dataset samples, and minimizing the risk of overfitting. The fine-tuned VGG16 architecture attained the best accuracy at 98.69% in terms of categorization in the proposed studies.

An effective hybrid optimization approach was used in [114] for the segmentation and classification of brain tumors. To improve categorization, the CNN features were extracted. The suggested chronological Jaya honey badger algorithm (CJHBA) was used to train the deep residual network (DRN), which was used to conduct the classification by using the retrieved features as input. The Jaya algorithm, the honey badger algorithm (HBA), and the chronological notion are all combined in the proposed CJHBA. Using BRATS-2018, the performance is assessed. The highest accuracy is 92.10%. A summary of MRI brain tumor classification using DL is provided in Table 7.

#### 5.3.4. Hybrid Techniques

Hybrid strategies use multiple approaches to achieve high accuracy, emphasizing each approach’s benefits while minimizing the drawbacks. The first method employed a segmentation technique to identify the part of the brain that was infected, and the second method for classification. Hybrid techniques are summarized in Table 8.

The proposed integrated SVM and ANN-based method for classification can be discovered in [115]. The FCM method is used to segment the brain MRI images initially, where the updated membership and k value diverge from the standard method. Two types of characteristics have been retrieved from segmented images to distinguish and categorize tumors. Using SVM, the first category of statistical features was used to differentiate between normal or abnormal brain MRI images. This SVM technique has an accuracy rate of 97.44%. Area, perimeter, orientation, and eccentricity were additional criteria used to distinguish between the tumor and various malignant stages I through IV. The tumor categories and stages of malignant tumors are classified through the ANN back-propagation technique. This suggested strategy has a 97.37% accuracy rate for categorizing tumor stages.

A hybrid segmentation strategy using ANN was suggested in [116] to enhance the brain tumor’s classification outcomes. First, the tumor region was segmented using skull stripping and thresholding. The segmented tumor was subsequently recognized using the canny algorithm, and the features of the identified tumor cell region were then used as the input of the ANN for classification; 98.9% accuracy can be attained with the provided strategy.

A system that can identify and categorize the different types of tumors as well as detect them in T1 and T2 image sequences was proposed by Ramdlon et al. [52]. Only the axial section of the MRI results, which are divided into three classifications (Glioblastoma, Astrocytoma, and Oligodendroglioma), are used for the data analysis using this method. Basic image processing techniques were used to identify the tumor region, including image enhancement, binarization, morphology, and watershed. Following the shape extraction feature segmentation, the KNN classifier was used to classify tumors; 89.5% of tumors were correctly classified.

Gurbina et al. [30] described the suggested integrated DWT and SVM classification methods. The initial segmentation of the brain MRI images was performed using Ostu’s approach. The DWT features were obtained from segmented images to identify and categorize tumors. Brain MRI images were divided into benign and malignant categories using an SVM classifier. This SVM method has a 99% accuracy rate.

The objective of the study in [117] is multilevel segmentation for effective feature extraction and brain tumor classification from MRI data. The authors used thresholding, the watershed algorithm, and morphological methods for segmentation after preprocessing the MRI image data. Through CNN, features are extracted, and SVM classed the tumor images as malignant or noncancerous. The proposed algorithm has an overall accuracy of 87.4%.

The classification of brain tumors into three types—glioblastoma, sarcoma, and metastatic—has been proposed by the authors of [118]. The authors first used FCM clustering to segment the brain tumor and then DWT to extract the features. PCA was then used to minimize the characteristics. Using six layers of DNN, categorization was completed. The suggested method displays 98% accuracy.

The method presented by Babu et al. [119] focused on categorizing and segmenting brain cancers from MRI images. Four processes compose the procedure: image denoising, segmentation of tumor, extracting features, and hybrid classification. They used the wavelet-based method to extract features after employing the thresholding process to remove tumors from brain MRI images. The final hybrid categorization was performed using CNN. The experiment’s findings showed that the approach had a segmentation accuracy of 95.23%, but the suggested optimized CNN had a classification accuracy of 99%.

Improved SVM was suggested as a novel algorithm by Ansari [120]. They recommended four steps for identifying and classifying brain tumors using MRI data: preprocessing, segmentation of images, extracting features, and image categorization. They segmented tumors using a fuzzy clustering approach and extracted key features using GLCM. In the classification stage, improved SVM was finally used. The suggested approach has an 88% accuracy rate.

A fully automated system for segmenting and diagnosing brain tumors was proposed by Farajzadeh et al. [121]. This is accomplished by first applying five distinct preprocessing techniques to an MR image, passing the images through a DWT, and then extracting six local attributes from the image. The processed images are then delivered to an NN, which subsequently extracts higher-order attributes from them. Another NN then weighs the features and concatenates them with the initial MR image. The hybrid U-Net is then fed with the concatenated data to segment the tumor and classify the image. For segmenting and categorizing brain tumors, they attained accuracy rates of 98.93% and 98.81%, respectively.

**Table 8 diagnostics-13-03007-t008:** Hybrid techniques.

Ref.	Year	Segmentation Method	Feature Extraction	Classifier	Accuracy
[115]	2017	FCM	shape and statistical	SVM and ANN	97.44% and 97.37%
[118]	2017	FCM	DWT and PCA	CNN	98.00%
[52]	2019	watershed	shape	KNN	89.50%
[30]	2019	Ostu’s	DWT	SVM	99.00%
[117]	2020	thresholding and watershed	CNN	SVM	87.4%.
[116]	2020	canny	GLCM and Gabor	ANN	98.90%
[119]	2023	thresholding	wavelet	CNN	99.00%
[120]	2023	fuzzy clustering	GLCM	Improved SVM	88.00%
[121]	2023	U-Net	DWT	CNN	98.93%

#### 5.3.5. Various Segmentation and Classification Methods Employing CT Images

Wavelet statistical texture features (WST) and wavelet co-occurrence texture features (WCT) were combined to segment brain tumors in CT images [122] automatically. After utilizing GA to choose the best texture features, two different NN classifiers were tested to segment the region of a tumor. This approach is shown to provide good outcomes with an accuracy rate of above 97%. Architecture of NN is shown in Figure 14.

For the segmentation and classification of cancers in brain CT images utilizing SVM with GA feature selection, a novel dominating feature extraction methodology was presented in [123]. They used FCM and K-means during the segmentation step and GLCM and WCT during the feature extraction stage. This approach is shown to provide positive results with an accuracy rate of above 98%.

An improved semantic segmentation model for CT images was suggested in [124]. Additionally, classification is used in the suggested work. In the suggested architecture, the semantic segmentation network, which has several convolutional layers and pooling layers, was used to first segment the brain image. Then, using the GoogleNet model, the tumor was divided into three groups: meningioma, glioma, and pituitary tumor. The overall accuracy achieved with this strategy was 99.6%.

A unique correlation learning technique utilizing CNN and ANN was proposed by Woniak et al. [125]. CNN used the support neural network to determine the best filters for the convolution and pooling layers. Consequently, the main neural classification improved efficiency and learns more quickly. Results indicated that the CLM model can achieve 96% accuracy, 95% precision, and 95% recall.

The contribution of image fusion to an enhanced brain tumor classification framework was examined by Nanmaran et al. [126], and this new fusion-based tumor categorization model can be more successfully applied to personalized therapy. A distinct cosine transform-based (DCT) fusion technique is utilized to combine MRI and SPECT images of benign and malignant class brain tumors. With the help of the features extracted from fused images, SVM, KNN, and decision trees were set to test. When using features extracted from fused images, the SVM classifier outperformed KNN and decision tree classifiers with an overall accuracy of 96.8%, specificity of 93%, recall of 94%, precision of 95%, and F1 score of 91%. Table 9 provides different segmentation and classification methods employing CT images.

## 6. Discussion

Most brain tumor segmentation and classification strategies are presented in this review. The quantitative efficiency of numerous conventional ML- and DL-based algorithms is covered in this article. Figure 15 displays the total number of publications published between 2010 and 2022 used in this review. Figure 16 displays the total number of articles published that perform classification, segmentation, or both.

Brain tumor segmentation uses traditional image segmentation methods like region growth and unsupervised machine learning. Noise, low image quality, and the initial seed point are its biggest challenges. The classification of pixels into multiple classes has been accomplished in the second generation of segmentation methods using unsupervised ML, such as FCM and K-means. These techniques are, nevertheless, quite noise sensitive. Pixel-level classification-based segmentation approaches utilizing conventional supervised ML have been presented to overcome this difficulty. Feature engineering, which extracts the tumor-descriptive pieces of information for the model’s training, is frequently used in conjunction with these techniques. Additionally, postprocessing helps further improve the results of supervised machine learning segmentation. Through the pipeline of its component parts, the deep learning-based approach accomplishes an end-to-end segmentation of tumors using an MRI image. These models frequently eliminate the requirement for manually built features by automatically extracting tumor descriptive information. However, their application in the medical domains is limited by the need for a big dataset for training the models and the complexity of understanding them.

In addition to the segmentation of the brain cancer region from the MRI scan, the classification of the tumor into its appropriate type is crucial for diagnosis and treatment planning, which in today’s medical practice necessitates a biopsy process. Several approaches that use shallow ML and DL have been put forth for classifying brain tumors. Type shallow ML techniques frequently include preprocessing, ROI identification, and feature extraction steps. Extracting descriptive information is a difficult task because of the inherent noise sensitivity associated with MRI image collection as well as differences in the shape, size, and position of tumor tissue cells. As a result, deep learning algorithms are currently the most advanced method for classifying many types of brain cancers, including astrocytomas, gliomas, meningiomas, and pituitary tumors. This review has covered several classifications of brain tumors.

The noisy nature of an MRI image is one of the most frequent difficulties in ML-based segmentation and classification of brain tumors. To increase the precision of brain tumor segmentation and classification models, noise estimation and denoising MRI images is a vital preprocessing operation. As a result, several methods, including the median filter [115], Wiener filter and DWT [30], and DL-based methods [117], have been suggested for denoising MRI images.

Large amounts of data are needed for DL models to operate effectively, but there need to be more datasets available. Data augmentation aids in expanding small datasets and creating a powerful generalized model. A common augmentation method for MRI images has yet to be developed. Although many methods have been presented by researchers, their primary goal is to increase the number of images. Most of the time, they ignore the connections between space and texture. An identical augmentation technique is required for comparative analysis to be conducted on its foundation.

## 7. General Problems and Challenges

Features are first manually extracted for ML, and are then fed into the ML-based differentiation system. Continuous variation within image classes makes utilizing ML-based algorithms for image classification challenging. Furthermore, the feature extraction methods’ usage of modern distance metrics makes it impossible to determine the similarity between two images.

Deep learning analyzes several parameters and optimizes them to extract and select features on its own. However, the system lacks intelligence in feature selection and typically pools, which reduces parameters and eliminates features that could be useful to the entire system.

Furthermore, DL models need data, and those data are coupled with millions or trillions of parameters. Therefore, enormous amounts of memory and GPU-based computers are required in the current environment. However, because of their high cost, these devices are not available to everyone. Consequently, many researchers need to create models that fit within their available budgets, which significantly impacts the quality of their study.

The noisy nature of an MRI image is one of the most frequent difficulties in ML-based brain tumor detection and classification. Preprocessing is necessary to remove all forms of noise from data and make it more suitable for the task at hand. Preprocessing difficulties exist in all the available datasets. However, the BRATS datasets have problems, such as motion artifacts and noise. There is no established preprocessing standard currently. People employ subpar application software, causing the image quality to decrease rather than improve.

### 7.1. Brain Cancer and Other Brain Disorders

#### 7.1.1. Stroke

Hemorrhagic strokes come from blood vessel injury or aberrant vascular structure, while ischemic strokes occur when the brain’s blood supply is cut off. Although the fact that strokes and brain tumors are two distinct illnesses, the connections associated with them have been studied [127]. 

They discovered that stroke patients are more likely than other cancer types to acquire brain cancer. Another intriguing conclusion of the study is that women between the ages of 40 and 60 and elderly stroke patients are more likely to acquire brain cancer.

#### 7.1.2. Alzheimer’s Disease

Short-term loss of memory is an initial symptom of Alzheimer’s disease (AD), a chronic neurodegenerative illness that may become worse over time as the disease progresses [108]. Despite AD and cancer being two distinct diseases, several studies have found a connection between them. According to the research, there is an inverse association between cancer and Alzheimer’s disease. They discovered that patients who had cancer had a 33% lower risk of Alzheimer’s disease than individuals who had not had cancer throughout the course of a mean follow-up of 10 years. Another intriguing finding of the study was that people with AD had a 61% lower risk of developing cancer.

## 8. Future Directions

The main applications of CADx systems are in educating and training; clinical practice is not one of them. CADx-based systems still need to be widely used in clinics. The absence of established techniques for assessing CADx systems in a practical environment is one cause of this. The performance metrics outlined in this study provide a helpful and necessary baseline for comparing algorithms, but because they are all so dependent on the training set, more advanced tools are required.

The fact that the image formats utilized to train the models were those characteristics of the AI research field (PNG) rather than those of the radiology field (DICOM, NIfTI) is noteworthy. Many of the articles analyzed needed authors with clinical backgrounds.

A different but related technical issue that may affect the performance of CADx systems in practice is the need for physician training on interacting with and interpreting the results of such systems for diagnostic decisions. This issue must be dealt with in all the papers included in the review. In terms of research project relevance and the acceptance of its findings, greater participation by doctors in the process may be advantageous.

## 9. Conclusions

A brain tumor is an abnormal growth of brain tissue that affects the brain’s ability to function normally. The primary objective in medical image processing is to find accurate and helpful information with the minimum possible errors by using algorithms. The four steps involved in segmenting and categorizing brain tumors using MRI data are preprocessing, picture segmentation, extracting features, and image classification. The diagnosis, treatment strategy, and patient follow-up can all be greatly enhanced by automating the segmentation and categorization of brain tumors. It is still difficult to create a fully autonomous system that can be deployed on clinical floors due to the appearance of the tumor and its irregular size, form, and nature. The review’s primary goal is to present the state-of-the-art in the field of brain cancer, which includes the pathophysiology of the disease, imaging technologies, WHO classification standards for tumors, primary methods of diagnosis, and CAD algorithms for brain tumor classifications using ML and DL techniques. Automating the segmentation and categorization of brain tumors using deep learning techniques has many advantages over region-growing and shallow ML systems. DL algorithms’ powerful feature learning capabilities are primarily to blame for this. Although DL techniques have made a substantial contribution, a general technique is still needed. This study reviewed 53 studies that used ML and DL to classify brain tumors based on MRI, and it examined the challenges and obstacles that CAD brain tumor classification techniques now face in practical application and advancement—a thorough examination of the variables that might impact classification accuracy. The MRI sequences and web address of the online repository for the dataset are among the publicly available databases that have been briefly listed in Table 4 and used in the experiments evaluated in this paper.

## Figures and Tables

**Figure 1 diagnostics-13-03007-f001:**
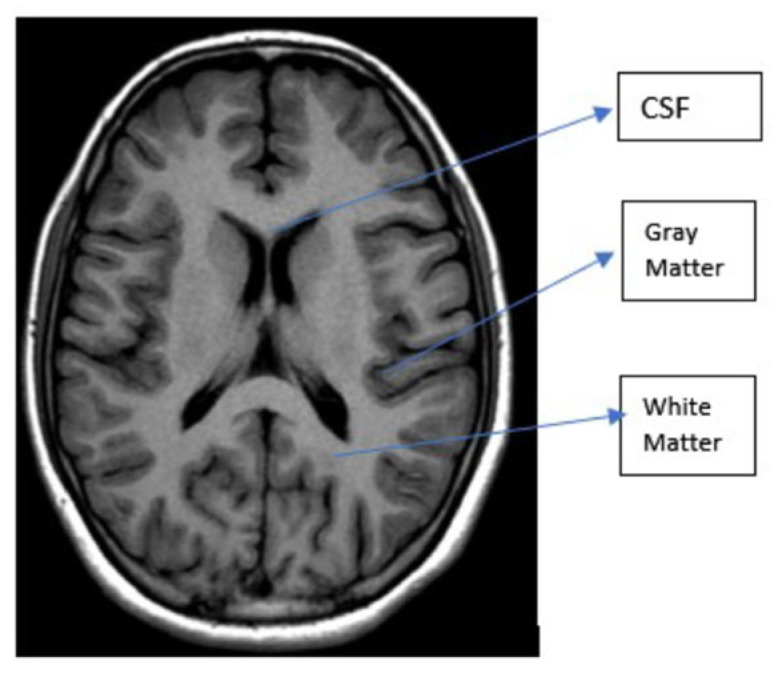
Healthy brain MRI image showing white matter (WM), gray matter (GM), and CSF [17].

**Figure 2 diagnostics-13-03007-f002:**
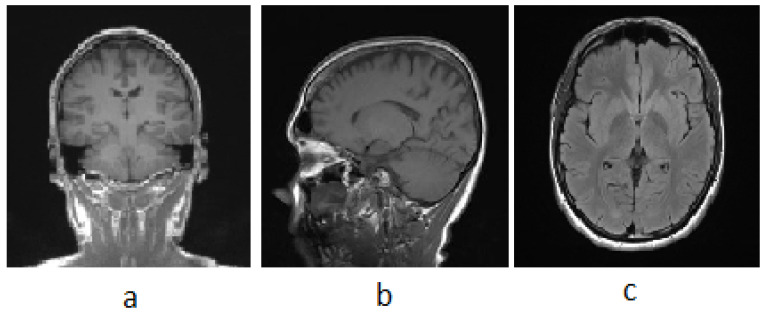
Fundamental MRI planes: (**a**) coronal, (**b**) sagittal, and (**c**) axial.

**Figure 3 diagnostics-13-03007-f003:**
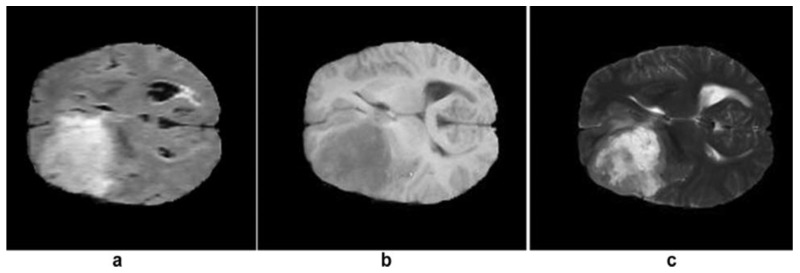
MRI brain tumor: (**a**) FLAIR image, (**b**) T1 image, and (**c**) T2 image [17].

**Figure 4 diagnostics-13-03007-f004:**
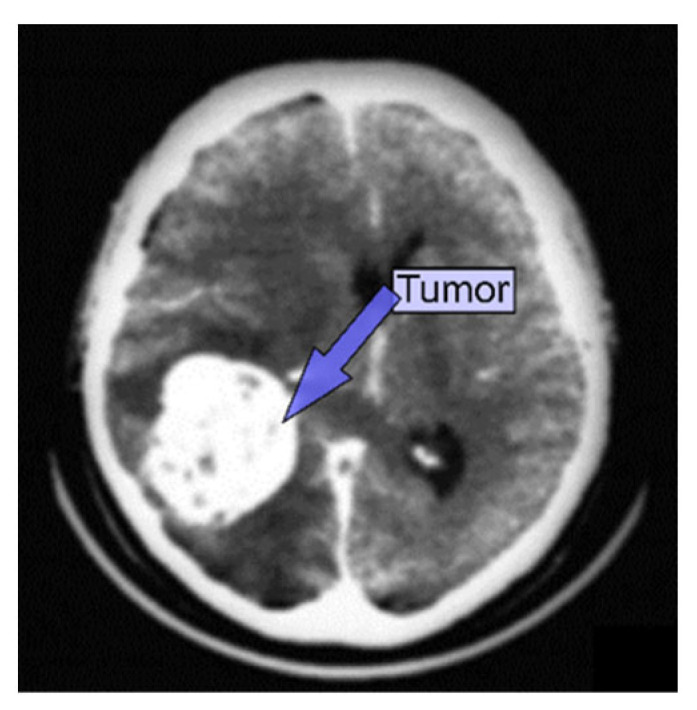
CT brain tumor.

**Figure 5 diagnostics-13-03007-f005:**
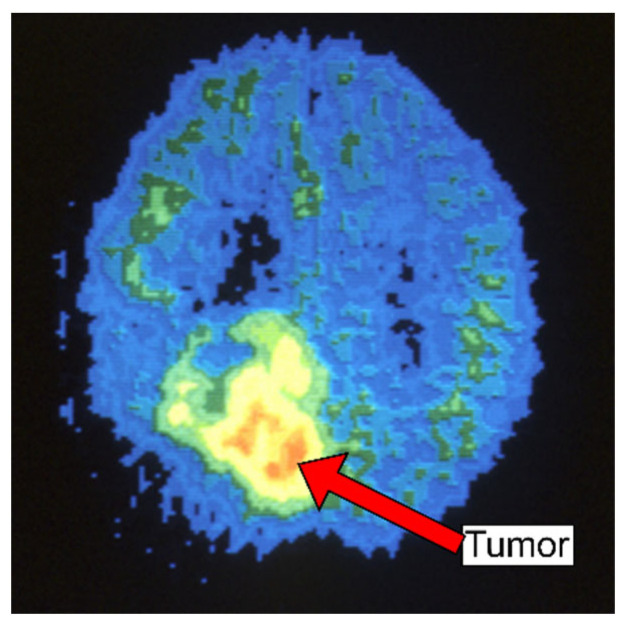
PET brain tumor.

**Figure 6 diagnostics-13-03007-f006:**
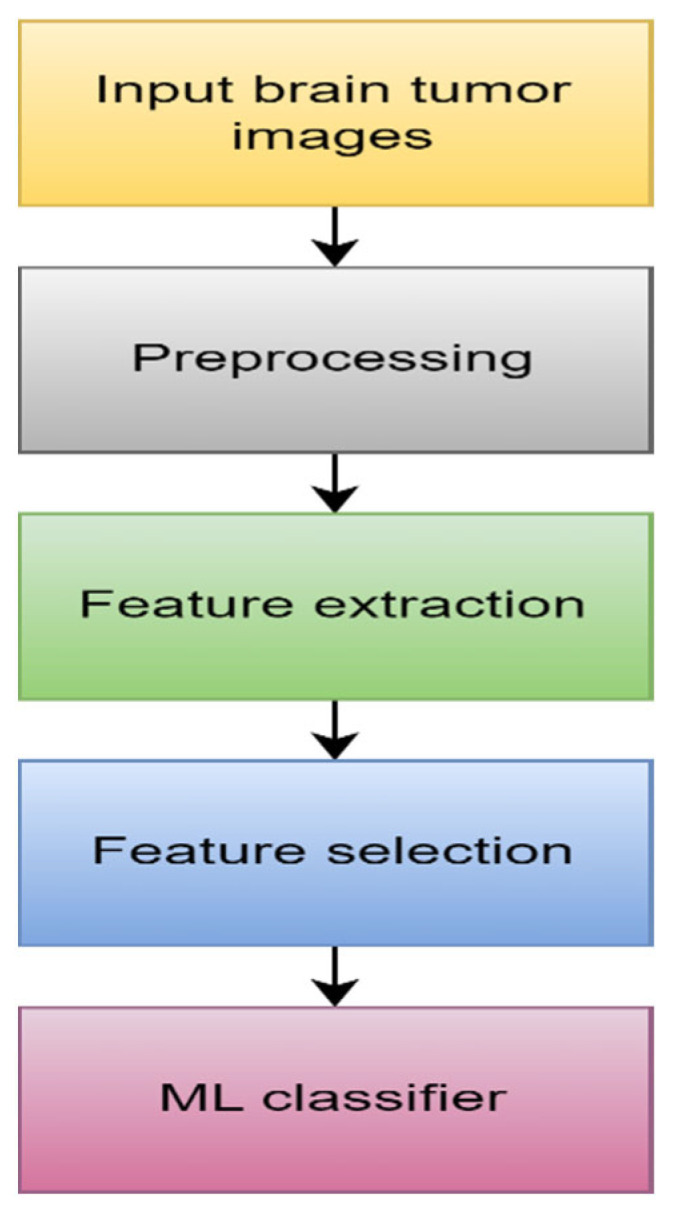
ML block diagram.

**Figure 7 diagnostics-13-03007-f007:**
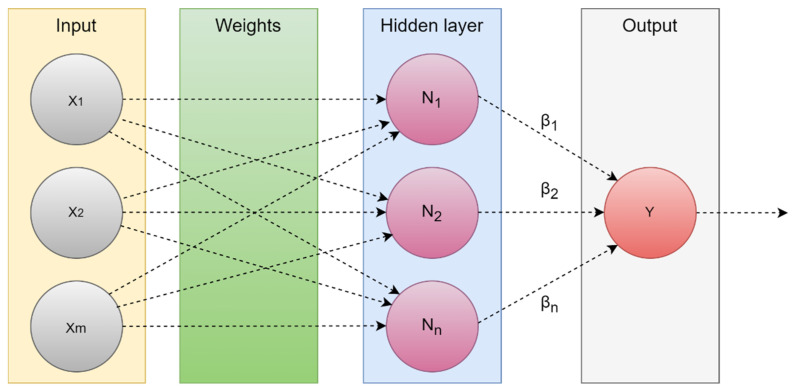
Extreme learning machine.

**Figure 8 diagnostics-13-03007-f008:**
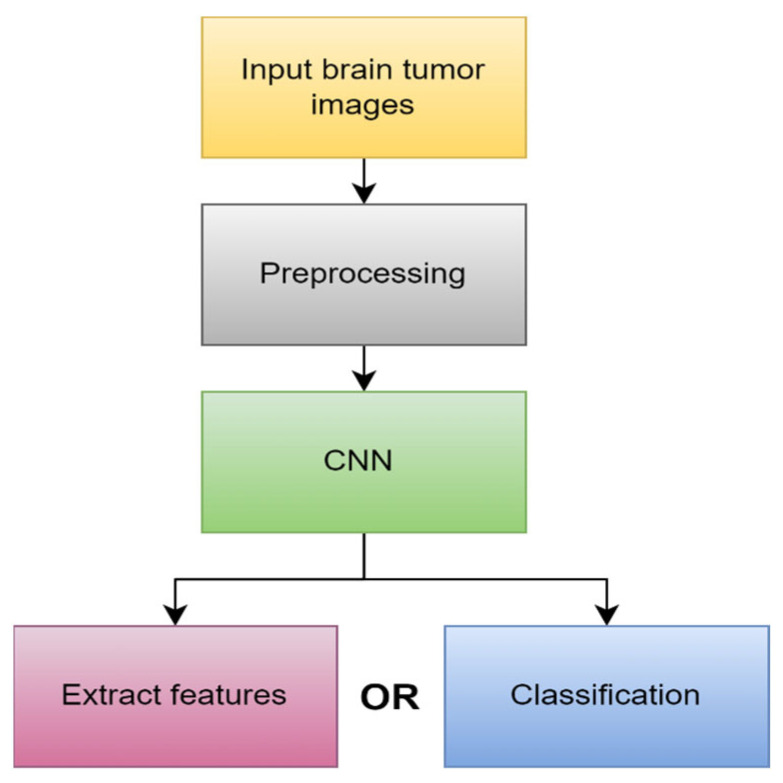
DL block diagram.

**Figure 9 diagnostics-13-03007-f009:**
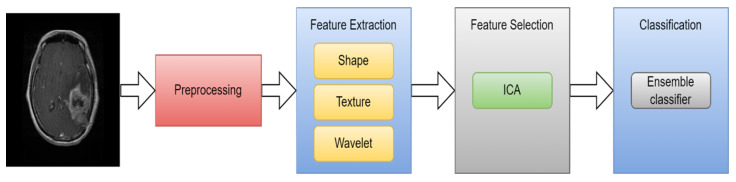
CAD method based on ensemble classifiers.

**Figure 10 diagnostics-13-03007-f010:**
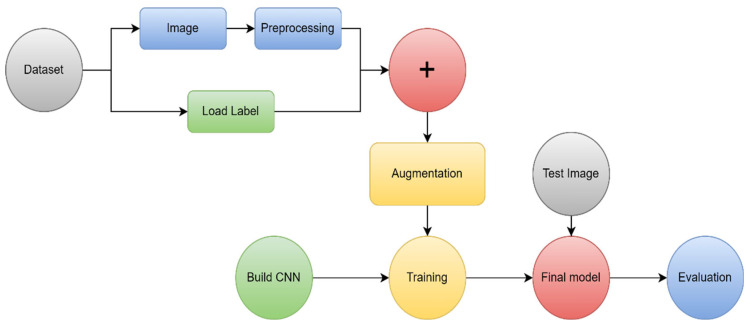
A block schematic showing the suggested approach. Reprinted (adapted) with permission from [7]. Copyright 2019 IEEE.

**Figure 11 diagnostics-13-03007-f011:**
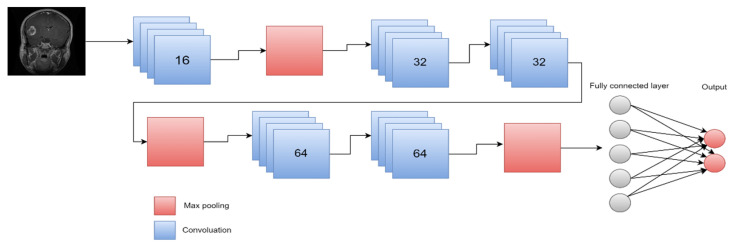
Proposed method. Reprinted (adapted) with permission from [102]. Copyright 2020 Mathematical Biosciences and Engineering.

**Figure 12 diagnostics-13-03007-f012:**
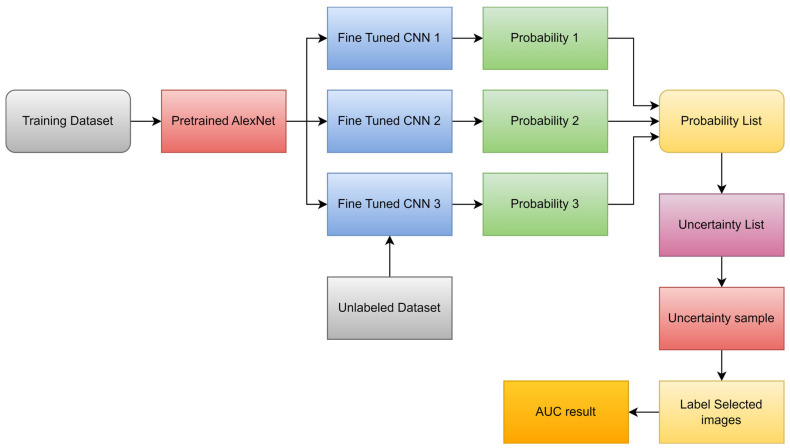
Workflow of the suggested active learning framework based on transfer learning. Reprinted (adapted) with permission from [104]. Copyright 2021 Frontiers in Artificial Intelligence.

**Figure 13 diagnostics-13-03007-f013:**
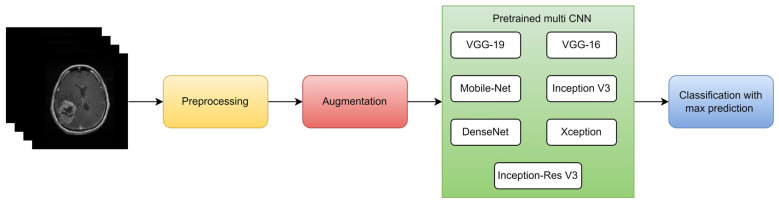
Proposed process for deep transfer learning. Reprinted (adapted) with permission from [106]. Copyright 2021 Indonesian Journal of Electrical Engineering and Computer Science.

**Figure 14 diagnostics-13-03007-f014:**
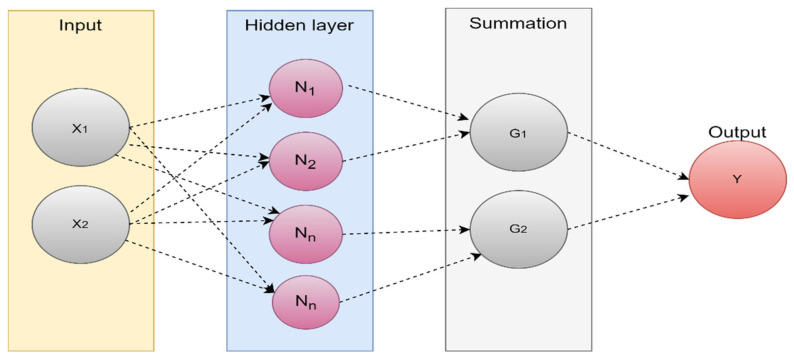
Architecture of NN.

**Figure 15 diagnostics-13-03007-f015:**
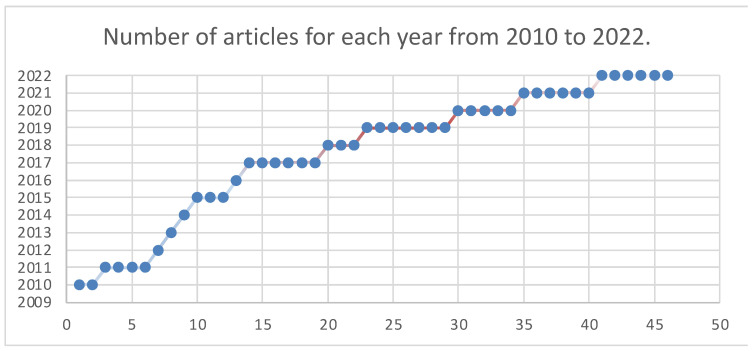
Number of articles published from 2010 to 2022.

**Figure 16 diagnostics-13-03007-f016:**
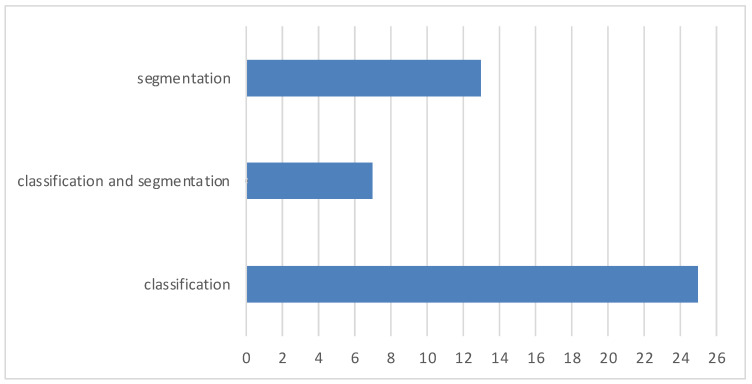
Number of articles published that perform classification, segmentation, or both.

**Table 1 diagnostics-13-03007-t001:** Types of brain tumors.

Types of Tumors Based on	Type	Comment
Nature	Benign	Less aggressive and grows slowly
	Malignant	Life-threatening and rapidly expanding
Origin	Primary tumor	Originates in the brain directly
	Secondary tumor	This tumor develops in another area of the body like lung and breast before migrating to the brain
Grading	Grade I	Basically, regular in shape, and they develop slowly
	Grade II	Appear strange to the view and grow more slowly
	Grade III	These tumors grow more quickly than grade II cancers
	Grade IV	Reproduced with greater rate
Progression stage	Stage 0	Malignant but do not invade neighboring cells
	Stage 1	Malignant and quickly spreading
	Stage 2	
	Stage 3	
	Stage 4	The malignancy invades every part of the body

**Table 2 diagnostics-13-03007-t002:** Properties of various MRI sequences.

	T1	T2	Flair
White Matter	Bright	Dark	Dark
Gray Matter	Gray	Dark	Dark
CSF	Dark	Bright	Dark
Tumor	Dark	Bright	Bright

**Table 3 diagnostics-13-03007-t003:** Performance equation.

Parameter	Equation
ACC	(TP+TN)/(TP+FN+FP+TN)
SEN	TP/(TP+FN)
SPE	TN/(TN+FP)
PR	TP/(TP+FP)
F1_SCORE	2∗PR∗SEN/(PR+SEN)
DCS	2∗TP/(2∗TP+FP+FN
Jaccard	TP/(TP+FP+FN)

**Table 4 diagnostics-13-03007-t004:** Summary of the dataset.

Dataset	MRI Sequences	Source
BRATS	T1, T2, FLAIR	[73]
RIDER	T1, T2, FLAIR	[74]
Harvard	T2	[75]
TCGA	T1, T2, FLAIR	[76,77]
Figshare	T1	[78]
IXI	T1, T2	[79]

**Table 5 diagnostics-13-03007-t005:** MRI brain tumor segmentation.

Ref.	Scan	Year	Technique	Method	Performance Metrics	Result
[80]	MRI	2010	region-based	FCM	Acc	93.00%
[81]	MRI	2011	region-based	FCM	Jaccard	83.19%
[82]	MRI	2012	NN	LBP with SVM	DSC	69.00%
[69]	MRI	2016	DL	CNN	DSC	88.00%
[84]	MRI	2017	NN	GLCM with SVM	DSC	86.12%
[38]	MRI	2017	NN	LBP with RF	Jaccard and DSC	87% and 93%
[85]	MRI	2018	region-based	FCM	Acc	98.00%
[83]	MRI	2018	region-based	FCM and k-mean	Acc	91.94%
[68]	MRI	2019	DL and NN	CNN with SVM	DSC	88.00%
[86]	MRI	2019	DL	Two-path CNN	DSC	89.20%
[87]	MRI	2019	DL	semantic	Acc	88.20%
[88]	MRI	2021	DL	semantic	IoU	91.72%
[89]	MRI	2022	DL	MRA-UNet	DSC	98.18%
[90]	MRI	2023	region-based	Fuzzy Otsu Threshold	Acc	94.37%

**Table 7 diagnostics-13-03007-t007:** MRI brain tumor classification using DL.

Ref.	Scan	Year	Technique	Method	Result	Performance Metrics
[101]	MRI	2015	DL	Custom-CNN	96.00%	Acc
[7]	MRI	2019	DL	Custom-CNN	98.70%	Acc
[102]	MRI	2020	DL	VGG-16, Inception-v3, ResNet-50	96% 75% 89%	Acc
[103]	MRI	2021	DL	AlexNet, GoogleNet, SqueezeNet	97.10%	Acc
[104]	MRI	2021	DL	Custom-CNN	82.89%	ROC
[105]	MRI	2018	DL	AlexNet	90.90%	Test acc
[106]	MRI	2021	DL	multi-CNN structure	98.67% 98.06% 98.33% 98.06%	precision, F1 score, precision, sensitivity
[107]	MRI	2022	DL	EfficientNetB0	98.80%	Acc
[70]	MRI	2022	DL	ResNet18	88.00%	AUC
[108]	MRI	2022	DL	Custom-CNN	98.70%	Acc
[109]	MRI	2022	DL	Custom-CNN	95.75%	Acc
[110]	MRI	2022	DL	Gaussian-CNN	99.80%	Acc
[111]	MRI	2020	DL	seven-layer CNN	97.52%	Acc
[112]	MRI	2021	DL	Alexnet	100.00%	Acc
[113]	MRI	2019	DL	VGG16	98.69%	Acc
[114]	MRI	2023	DL	CNN	92.10%	Acc

**Table 9 diagnostics-13-03007-t009:** Various segmentation and classification methods employing CT images.

Ref.	Year	Type	Segmentation	Feature Extraction	Feature Selection	Classification	Result
[122]	2011	CT	NN	WCT and WST	GA	-	97.00%
[123]	2011	CT	FCM and k-mean	GLCM and WCT	GA	SVM	98.00%
[124]	2020	CT	Semantic	-	-	GoogleNet	99.60%
[125]	2021	CT	-	-	-	CNN	96.00%
[126]	2022	SPECT/MRI	-	DCT	-	SVM	96.80%

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
