# Peer review of "A Review of Recent Advances in Brain Tumor Diagnosis Based on AI-Based Classification"

_diagnostics, 2023, doi:10.3390/diagnostics13183007_

Round 1

Reviewer 1 Report

1.Unified capitalization of keywords

2. The abstract needs to describe the work done in this paper to make it easier for others to understand

3. The authors have made a good summary, but it is recommended to add the common utilized datasets to facilitate readers to quickly get started

4. In the Discussion, the authors need to point out the current research issues and future research content.

Grammar needs polishing

Author Response

Reviewer 1

1.Unified capitalization of keywords

The capitalization of keywords was unified as suggested. 

2. The abstract needs to describe the work done in this paper to make it easier for others to understand.

As suggested, the work done in this paper was added in the abstract. Lines 21-28.

3. The authors have made a good summary, but it is recommended to add the common utilized datasets to facilitate readers to quickly get started.

As suggested, the common utilized dataset was added. Lines 494-499. Table 4.

4. In the Discussion, the authors need to point out the current research issues and future research content.

As suggested, general problems and challenges were added. Lines 958-1014.

5. Grammar needs polishing

A native speaker carefully reviewed the manuscript and skillfully corrected any grammar mistakes as recommended.

Reviewer 2 Report

The review topic should be relevant to specific diagnostic techniques. The given review is just an overview of the brain tumor diagnosis and segmentation. The given review consists of two topics including brain tumor diagnosis and segmentation techniques. Though these two topics are relevant to each other but writing a review on two topics seems to be inappropriate. Or authors must address the relevancy of segmentation on brain tumor diagnosis.

1.       Authors have performed detailed literature review on deep learning techniques for image classification and segmentations. Author have cited and presented the published figures in the paper, but copyrights statements are missing.

2.       It is suggested to change the title of review article to be more specific. The author can make some changes based on suggested example: “A review of recent advances in brain tumor diagnosis based on imaging techniques”, or “A review of recent advances in brain tumor diagnosis based on AI based classification”.

3.       Page # 3, Line 102, author didn’t provide any information about the FMRI.

4.       The most of figures didn’t provide any informative details. Resolution of the images are very low and whether authors have taken permission or copyrights to publish the figures?

5.       Page # 6, “A nuclear imaging examination called a SPECT”, whereas SPECT abbreviation is single-photon emission computed tomography. The description is not clear.

6.       Line 475, typo error, the space is missing: (2010 to 2022)  instead (2010to 2022).

7.       Line 782, “Figure 21 illustrated the proposed method”. But there is no Fig. 21 in the paper.

8.       Line 860, “NN classifiers as shown in figure 22 were tested to segment the region of a tumor.” But there is no Fig. 22 in the paper.

9.       Line 905, authors have mentioned Figure 23, but there is no such figure.

Author Response

Reviewer 2

The review topic should be relevant to specific diagnostic techniques. The given review is just an overview of the brain tumor diagnosis and segmentation. The given review consists of two topics including brain tumor diagnosis and segmentation techniques. Though these two topics are relevant to each other but writing a review on two topics seems to be inappropriate. Or authors must address the relevancy of segmentation on brain tumor diagnosis.

Thank you for your suggestion. As suggested, the relevancy of segmentation on brain tumor diagnosis was added. Lines 362-366.

1. Authors have performed detailed literature review on deep learning techniques for image classification and segmentations. Author have cited and presented the published figures in the paper, but copyrights statements are missing.

The figures were cited and were regenerated

2. It is suggested to change the title of review article to be more specific. The author can make some changes based on suggested example: “A review of recent advances in brain tumor diagnosis based on imaging techniques”, or “A review of recent advances in brain tumor diagnosis based on AI based classification”.

As suggested, the topic was updated to “A review of recent advances in brain tumor diagnosis based on AI based classification”.

3. Page # 3, Line 102, author didn’t provide any information about the FMRI.

FMRI was added in lines 156-159.

4. The most of figures didn’t provide any informative details. Resolution of the images are very low and whether authors have taken permission or copyrights to publish the figures?

All figures were re generated and cited and additional information were added.

5. Page # 6, “A nuclear imaging examination called a SPECT”, whereas SPECT abbreviation is single-photon emission computed tomography. The description is not clear.

SPECT was revised lines 190-191.

6. Line 475, typo error, the space is missing: (2010 to 2022)  instead (2010to 2022).

Revised as suggested. Line 487.

7. Line 782, “Figure 21 illustrated the proposed method”. But there is no Fig. 21 in the paper.

Revised as suggested. Lines 796-798.

8. Line 860, “NN classifiers as shown in figure 22 were tested to segment the region of a tumor.” But there is no Fig. 22 in the paper.

Revised as suggested. Lines 874-878.

9. Line 905, authors have mentioned Figure 23, but there is no such figure.

Revised as suggested. Lines 911-915.

Reviewer 3 Report

1- The motivation of this work is not clear. What motivates you to perform this task?

2-  This area is rapidly evolving, and new papers have been published. Therefore, some state-of-the-art papers should be taken into account are:       https://www.sciencedirect.com/science/article/pii/S0020025522007332    https://www.sciencedirect.com/science/article/abs/pii/S0306987720308689 

3- Include a table of existing studies that should highlight the strengths and weaknesses of those methods.

4- Why Deep learning based methodologies are better than traditional machine learning methods i.e. SVM, KNN, ANN. Please explain this in detail. 

5- Since this is a review paper add more SOTA works for instance, 2022 and 2023. 

6- The conclusion section should be further enhanced.

7- Add more detail to the caption of Figure 1,  Figure 2, Figure 3, and Figure 14, 

8- Also the images resolutions are very low and are not clearly visible on print form. Please improve that. 

9- Typos:

There are too many typos. Please correct that. Figures numbers are not correct. E.g., Figure 14. It should be corrected. 

10- The authors should also run the manuscript through a grammar checker like Grammarly to address any language or grammatical errors. Finally, the authors should ensure that all references cited in the manuscript are up-to-date and relevant to the research topic.

The authors should also run the manuscript through a grammar checker like Grammarly to address any language or grammatical errors. Finally, the authors should ensure that all references cited in the manuscript are up-to-date and relevant to the research topic.

Author Response

Reviewer 2

1- The motivation of this work is not clear. What motivates you to perform this task?

The motivation is added as suggested in lines 72-79.

 2-  This area is rapidly evolving, and new papers have been published. Therefore, some state-of-the-art papers should be taken into account are:       https://www.sciencedirect.com/science/article/pii/S0020025522007332    https://www.sciencedirect.com/science/article/abs/pii/S0306987720308689

Added as suggested in lines 602-614.

3- Include a table of existing studies that should highlight the strengths and weaknesses of those methods.

We appreciate the reviewer input. However, adding such details will require more time to revise. from our point of view not adding the table won’t harm the quality of our paper.

4- Why Deep learning based methodologies are better than traditional machine learning methods i.e. SVM, KNN, ANN. Please explain this in detail.

Added as suggested in lines 682-688.

5- Since this is a review paper add more SOTA works for instance, 2022 and 2023.

Added as suggested.

6- The conclusion section should be further enhanced.

The conclusion was revised for enhancement as suggested.

7- Add more detail to the caption of Figure 1,  Figure 2, Figure 3, and Figure 14,

Added.

8- Also the images resolutions are very low and are not clearly visible on print form. Please improve that.

We tried to improve the resolutions of the images as suggested.

9- Typos:

There are too many typos. Please correct that. Figures numbers are not correct. E.g., Figure 14. It should be corrected.

Corrected.

10- The authors should also run the manuscript through a grammar checker like Grammarly to address any language or grammatical errors. Finally, the authors should ensure that all references cited in the manuscript are up-to-date and relevant to the research topic.

The manuscript was checked for any language or grammatical errors.

Round 2

Reviewer 2 Report

Figure numbers are wrong. Figure 9 onwards, author has started again from Fig 1. Figures are randomly arranged in the manuscript.

There must be copyright statement for figures taken from other sources.

Discussion section should be concise. No need for adding subheading and figures in Discussion section.

Author Response

Reviewer 3

Figure numbers are wrong. Figure 9 onwards, author has started again from Fig 1. Figures are randomly arranged in the manuscript.

There must be copyright statement for figures taken from other sources.

Discussion section should be concise. No need for adding subheading and figures in Discussion section

Figure numbers were checked and corrected.

Statements were added.  all figures that have been cited have been updated to include copyright statements, whereas any figures without citations are original.

The subheadings were added based on the suggestions received by reviewers from the previous revision.

Reviewer 3 Report

The authors have carefully addressed all the comments. Thanks